# Quantum-Inspired Fully Complex-Valued Neutral Network for Sentiment Analysis

**Wei Lai** [1], **Jinjing Shi** [1,*] and **Yan Chang** [2]

1    School of Computer Science and Engineering, Central South University, Changsha 410083, China
2    Advanced Cryptography and System Security Key Laboratory of Sichuan Province, Chengdu 610025, China
*    Correspondence: shijinjing@csu.edu.cn

**Abstract:** Most of the existing quantum-inspired models are based on amplitude-phase embedding to model natural language, which maps words into Hilbert space. In quantum-computing theory, the vectors corresponding to quantum states are all complex values, so there is a gap between these two areas. Presently, complex-valued neural networks have been studied, but their practical applications are few, let alone in the downstream tasks of natural language processing such as sentiment analysis and language modeling. In fact, the complex-valued neural network can use the imaginary part information to embed hidden information and can express more complex information, which is suitable for modeling complex natural language. Meanwhile, quantum-inspired models are defined in Hilbert space, which is also a complex space. So it is natural to construct quantum-inspired models based on complex-valued neural networks. Therefore, we propose a new quantum-inspired model for NLP, ComplexQNN, which contains a complex-valued embedding layer, a quantum encoding layer, and a measurement layer. The modules of ComplexQNN are fully based on complex-valued neural networks. It is more in line with quantum-computing theory and easier to transfer to quantum computers in the future to achieve exponential acceleration. We conducted experiments on six sentiment-classification datasets comparing with five classical models (TextCNN, GRU, ELMo, BERT, and RoBERTa). The results show that our model has improved by 10% in accuracy metric compared with TextCNN and GRU, and has competitive experimental results with ELMo, BERT, and RoBERTa.

**Keywords:** quantum theory; sentiment analysis; machine learning; natural language processing

**MSC:** 68T07; 68T50

## 1. Introduction

A quantum-inspired model is a new neural-network model constructed by combining quantum-computing theory and deep-learning theory. It is a heuristic method proposed when the quantum hardware development is limited in the noisy intermediate-scale quantum (NISQ) era. It draws lessons from the ideas of quantum-computing theory and adopts classical computing methods. A quantum-inspired language model is based on the mathematical framework of quantum-computing theory to model natural language, which is inspired by the similarity between natural language and the quantum system, such as the polysemous phenomenon of language and superposition of particles, the change in language over time and space and the continuous evolution of particles over time, the semantic determination of language in a specific scene and the collapse of particles to a certain ground state after measurement. Compared with classical neural networks, quantum-inspired language models are more consistent with the characteristics of natural language, and have better interpretability and exponential acceleration potential [1]. Moreover, the quantum-inspired language model is defined in Hilbert space, which has a stronger representation ability.

In recent years, natural language processing (NLP), benefiting from the development of deep learning, has made significant progress in many fields, including sentiment analysis [2], question answering [3], text generation [4], and so on. Recurrent neural networks (RNNs), which can learn strong correlations from text sequences, are commonly used neural-network modules in NLP. Long short-term memory (LSTM) and gated recurrent units (GRU) are two widely used RNNs. However, gradient explosion and gradient disappearance are problems with RNNs that make it hard to build a deep neural network. Based on self-attention mechanisms and residual structure, Transformer [5] can focus on key information in text and remember previous knowledge, which is suitable for building deep neural networks. Currently, the main technology used in NLP is the pretraining language model. Common pretraining language models include BERT [6], RoBERTa [7], GPT-3 [8], etc. Using fine-tuning techniques, these pretraining models can serve as text encoders that are easily adaptable to various downstream tasks. At present, the research direction of NLP has begun to develop towards larger datasets and larger models, such as GPT-3, where the size of the training corpus has reached 750 GB, and the number of references is as high as 175 billion. Although computing power is increasing year by year, researchers still feel constrained in the face of such large datasets and models with more than 100 billion references.

Quantum computing is a brand new computing theory. It has been demonstrated that quantum computers have an exponential computational complexity acceleration advantage for some tasks. For example, the famous Shor algorithm [9] can complete integer prime factorization with polynomial complexity, which threatens the communication process based on the classical RSA encryption algorithm. At present, there are also some works on modeling natural language based on quantum-computing theory, which is mainly due to two reasons. First, human language and quantum systems share a lot of similarities, such as language ambiguity and quantum superposition state, language evolution and quantum state evolution. The second is that quantum computing has the potential advantage of exponential speedup, which is very attractive for current pretraining methods that require significant resources to train networks. There have been some studies on quantum natural-language processing (QNLP). For example, Bob Coecke et al. proposed DisCoCat [10] and lambeq [11], which encode natural languages into string diagrams and then encode them into quantum circuits. Through parameterized quantum-circuit learning, text classification tasks can be realized. However, the experimental progress of quantum machine learning is still in its preliminary stage due to the limitations of quantum bits and error correction capabilities in quantum-computing devices. Therefore, if running on a real quantum computer, we can only handle a dataset of 100 sentences containing more than 10 words [12]. Some researchers use the mathematics behind quantum-computing theory to build models that look like quantum computers. These models do not need to run on real quantum computers, but just draw on relevant concepts in quantum-computing theory to help with natural-language modeling, so they are not bound by the development of hardware.

The quantum-inspired model aims to simulate natural language using the theory of quantum computation, analogizing natural language to a quantum system, and using a classical neural-network model to simulate this process. However, the existing quantum language models, such as NNQLM [13], CNM [14], etc., simulate the construction of quantum states by amplitude-phase embedding and obtaining the complex-valued representation of quantum states using Euler's formula. In fact, it is also feasible to directly use complex-valued neural networks to construct quantum-inspired models. There are many researchers working on complex-valued neural networks. Trabelsi et al. [15] proposed a deep complex-valued convolutional network that has demonstrated good performance in image classification, music transcription, and speech spectrum prediction.

In this work, our motivation is to learn how to design a quantum-inspired model that is more suitable for transfer to quantum computers to reduce the complexity of future language models. Under the condition that the development of existing quantum hardware be limited, the mathematical framework of quantum computing is used to realize the

quantum-inspired model so that it can run on a classical computer. However, previous quantum-inspired language models are rarely based on complex-valued neural networks. These works use real-valued neural networks that are not suitable for transfer to quantum computers defined in Hilbert space. Therefore, we propose a quantum-inspired fully complex-valued neural network, ComplexQNN, and use it to solve sentiment-classification tasks. At the end of the experiment, we discuss the advantages of ComplexQNN compared with classical neural-network models.

In summary, as the existing quantum-inspired models are rarely based on complex-valued neural networks, we proposed a new quantum-inspired fully complex-valued neural network, ComplexQNN. Our contributions are as follows:

- Based on quantum computation theory and complex-valued neural networks, we propose the theory and architecture of the ComplexQNN.
- We introduce the detailed modules of the ComplexQNN fully based on complex-valued neural networks, including a complex-valued embedding layer, a quantum encoding layer, and a measurement layer.
- The ComplexQNN is evaluated with six sentiment-classification datasets, including binary classification and multi-classification. We adopt two metrics—accuracy and F1-score—to evaluate our model and compare it with five classical neural models (TextCNN, GRU, ELMo, BERT, and RoBERTa). The experimental results show that the ComplexQNN has 10% improved accuracy compared with TextCNN and GRU, and has competitive experimental results with ELMo, BERT, and RoBERTa.

The rest of the paper is organized as follows: Section 2 reviews the literature and summarizes related works; Section 3 describes the materials and methods used throughout the study; Section 4 explains and discusses our experimental results; and Section 5 describes our conclusions.

## 2. Related Works

In this section, we will introduce the knowledge related to quantum-inspired complex-valued neural networks, including quantum computing, complex-valued neural networks, and the research progress of quantum-inspired neural networks.

### 2.1. Preliminary

Quantum computing is a new way of computing based on the idea of quantum mechanics. Classical computers can simulate quantum computers, but not very efficiently. Some quantum algorithms have been proposed to prove that quantum computers have the ability to accelerate classical computational problems. Peter Shor proposed in 1994 that quantum computers could solve the prime-factor problem of finding integers and solve the so-called discrete logarithm problem [9] . Lov Grover proved in 1995 that quantum computers can speed up the search problem in unstructured search spaces [16] . Wang et al. [17] proposed in 2021 a quantum AdaBoost algorithm with a quadratic speedup. Apers et al. [18] proposed in 2022 a continuous-time quantum walks (CTQWs) search algorithm, which achieves a general quadratic speedup over classical random walks on an arbitrary graph. Huang et al. [19] proposed a quantum principal component analysis that achieved almost four orders of magnitude of reduction over the best-known classical lower bounds. We will introduce some basic concepts in quantum computing, such as quantum state, quantum system, quantum state evolution, and quantum measurement (The most of these basic concepts are from Nielsen et al.'s book "Quantum computation and quantum information" [20]).

#### 2.1.1. Quantum State

In classical computing, a bit is used to represent two different states, such as 0 and 1. In quantum computing, a qubit is the basic unit, and the Dirac symbol is usually used to describe a qubit [20], such as $|0\rangle$ and $|1\rangle$. Moreover, a qubit can be in a superposition

state of $|0\rangle$ and $|1\rangle$. A total of $2^n$ data can be stored in $n$ qubits, which has the advantage of parallel computing and can bring exponential improvements to classical methods.

A qubit can be a linear combination of ground states $|0\rangle$ and $|1\rangle$, such as

$$|\psi\rangle = \alpha|0\rangle + \beta|1\rangle, \tag{1}$$

where $|\psi\rangle$ is often called a superposition state, and both $\alpha$ and $\beta$ are complex-valued numbers. In addition, it is impossible to obtain all the information about an unknown qubit completely. You can obtain $|0\rangle$ or $|1\rangle$ by measuring, where $|\alpha|^2$ is the probability of obtaining $|0\rangle$ and $|\beta|^2$ is the probability of obtaining $|1\rangle$, and satisfy $|\alpha|^2 + |\beta|^2 = 1$. Such as $\frac{1}{\sqrt{2}}|0\rangle + \frac{1}{\sqrt{2}}|1\rangle$, there is 50% of obtaining $|0\rangle$ and 50% of obtaining $|1\rangle$. Therefore, the equation can also be expressed as

$$|\psi\rangle = e^{i\gamma}(\cos\frac{\theta}{2}|0\rangle + e^{i\varphi}\sin\frac{\theta}{2}|1\rangle), \tag{2}$$

where $\gamma, \theta, \varphi$ are all real numbers. Additionally, $e^{i\gamma}$ can be omitted since it does not have any observable effect.

### 2.1.2. Quantum System

Two qubits have four ground states, and a pair of qubits can also be in the superposition of these four ground states, such as

$$|\psi\rangle = \alpha_{00}|00\rangle + \alpha_{01}|01\rangle + \alpha_{10}|10\rangle + \alpha_{11}|11\rangle, \tag{3}$$

where $\alpha_{ij}(i, j \in \{0, 1\})$ is called the amplitude. In a two-qubit system, you can measure only one of the qubits, such as the second bit. Assuming that the measurement result is 1, the measured state $|\psi\rangle$ will collapse to

$$|\psi'\rangle = \frac{\alpha_{01}|01\rangle + \alpha_{11}|11\rangle}{\sqrt{|\alpha_{01}|^2 + |\alpha_{11}|^2}}. \tag{4}$$

The factor $\sqrt{|\alpha_{01}|^2 + |\alpha_{11}|^2}$ is used for normalization. The Bell state $\frac{|00\rangle + |11\rangle}{\sqrt{2}}$ is a very important two-quantum state because it satisfies a property: the measurement result of two qubits is always the same. It is an indispensable part of quantum teleportation and ultra-dense coding. Consider $n$-qubit system, where the ground state is $|x_1 x_2 \ldots x_n\rangle$ and has $2^n$ amplitudes. Compared with classical systems, $n$ qubit systems have an exponential increase in storage and computation.

### 2.1.3. Quantum State Evolution

A quantum computer consists of quantum circuits and quantum gates, which are used to process quantum information. In a classical computer, logic gates are used to process classical information, such as NOT gates can change the state of a bit, changing 0 to 1 and 1 to 0. Similarly, there is a quantum NOT gate $X$ in a quantum computer, which can be expressed as

$$X = \begin{bmatrix} 0 & 1 \\ 1 & 0 \end{bmatrix}. \tag{5}$$

The outcome of the NOT gate of quantum state $|\psi\rangle = \alpha|0\rangle + \beta|1\rangle = [\alpha, \beta]^T$ is

$$X \begin{bmatrix} \alpha \\ \beta \end{bmatrix} = \begin{bmatrix} \beta \\ \alpha \end{bmatrix}. \tag{6}$$

A single-qubit quantum gate can be given by a $2 \times 2$ matrix. Since the qubit has a normalization condition, the corresponding matrix of the single-qubit gate must satisfy the

unitary property $U^\dagger U = I$, where $U^\dagger$ is the conjugate transpose of $U$, and $I$ is the identity matrix of $2 \times 2$ [20]. The Hadamard gate is another single-qubit gate that is often used. It is described by a matrix as

$$H = \frac{1}{\sqrt{2}} \begin{bmatrix} 1 & 1 \\ 1 & -1 \end{bmatrix}. \tag{7}$$

It changes $|0\rangle$ to the intermediate state $|+\rangle = (|0\rangle + |1\rangle)/\sqrt{2}$. Similarly, change $|1\rangle$ to the intermediate state $|-\rangle = (|0\rangle - |1\rangle)/\sqrt{2}$. In addition, any quantum computation on any number of qubits can be produced with a finite set of gates.

### 2.1.4. Quantum Measurement

By measuring quantum the state $|\psi\rangle = \alpha|0\rangle + \beta|1\rangle$, $|0\rangle$ and $|1\rangle$ can be obtained. In fact, this is to take the ground state as $|0\rangle$ and $|1\rangle$. You can also choose $|+\rangle$ and $|-\rangle$, the quantum state $|\psi\rangle$ can be re-expressed as

$$|\psi\rangle = \alpha|0\rangle + \beta|1\rangle = \frac{\alpha + \beta}{\sqrt{2}}|+\rangle + \frac{\alpha - \beta}{\sqrt{2}}|-\rangle. \tag{8}$$

So, there is a probability of $\frac{|\alpha+\beta|^2}{2}$ of obtaining $|+\rangle$ and $\frac{|\alpha-\beta|^2}{2}$ of obtaining $|-\rangle$. More generally, take a set of measurement operators $\{M_k\}$ to measure the quantum system $|\psi\rangle$—the probability of the measurement result being $k$ is

$$p(k) = \langle\psi|M_k^\dagger M_k|\psi\rangle, \tag{9}$$

where $^\dagger$ means conjugate transpose. The state of the system after measurement is

$$|\psi'\rangle = \frac{M_k|\psi\rangle}{\sqrt{\langle\psi|M_k^\dagger M_k|\psi\rangle}}, \tag{10}$$

where the operators satisfies $\sum_k M_k^\dagger M_k = I$. Therefore, the sum of the probabilities of all measurement results is 1 and it is described as

$$\sum_k \langle\psi|M_k^\dagger M_k|\psi\rangle = \sum_k p(k) = 1. \tag{11}$$

At present, quantum computers are facing problems such as high R&D costs, susceptibility to noise, and difficulty in exiting the experimental environment [21]. It is very difficult to realize a universal quantum computer, and it still needs a long period of in-depth research [22]. Meanwhile, the design of quantum algorithms needs to address two challenges [20]. First, quantum computing uses qubits instead of classical bits, so quantum algorithms need to consider how to use the superposition and entanglement properties of qubits to achieve parallel computing. Second, quantum algorithms need to be more efficient than existing classical algorithms. Otherwise, there is no need to use a quantum computer. This is difficult to achieve with current hardware constraints, and most quantum algorithms can only be theoretically proven to have an acceleration advantage. Therefore, many researchers have begun to study hybrid quantum classical algorithms and quantum-inspired algorithms. These methods do not rely on expensive quantum devices and can be run directly on a classical computer.

### 2.2. Complex Neural Network

Most of the existing deep-learning technologies are based on real-valued operations and representations [13,14,23]. In fact, complex numbers may have richer representation capabilities. Some works have proved that complex-valued neural networks have some unique advantages [15]: it is possible to achieve easier optimization [24], better generaliza-

tion features [25], faster learning [26], and noise-resistant memory mechanisms [27]. In 2018, Chiheb Trabelsi et al. [15] proposed a deep complex-valued neural network, proposed key modules for training complex-valued neural networks such as complex-valued batch normalization and complex-valued weight initialization, and also proposed a complex-valued convolutional neural-network architecture, and passed experiments on image classification, music transcription, and speech spectrum prediction were conducted to validate the effectiveness of the proposed networks.

The following introduces the principle of complex-valued neural networks, including complex-valued linear layers, complex-valued CNNs and complex-valued RNNs.

### 2.2.1. Complex-Valued Linear Layers

The linear layer is also called a fully connected layer [28]. Each neuron in the linear layer is connected to all neurons in the previous layer, which is the most common network structure in neural networks. Its calculation process is described by

$$f = WX + b, \tag{12}$$

where $W$ represents the weight matrix in the network, and $b$ represents the bias in the network layer. Trabelsi et al. [15] implemented complex-valued linear layers based on the PyTorch (https://pytorch.org/, accessed on 3 October 2016) library. The complex-valued linear layer uses two real-valued linear layers to calculate the real part and imaginary part, respectively, and obtain new real and imaginary parts based on the complex-valued calculation principle. Specifically, the calculation process is as follows:

$$f_r(X) = W_r X + b_r, \tag{13}$$

$$f_i(X) = W_i X + b_i, \tag{14}$$

$$f_c(X) = f_r(X_r) - f_i(X_i) + \mathbf{i} \times [f_r(X_i) + f_i(X_r)] \tag{15}$$

$$= W_r X_r - W_i X_i + b_r - b_i + \mathbf{i} \times [W_r X_i + W_i X_r + b_r + b_i], \tag{16}$$

where $\mathbf{i}$ is the imaginary part. The complex-valued linear layer is the basic module of the complex-valued neural network, and both the complex-valued CNNs and the complex-valued RNNs rely on this module.

### 2.2.2. Complex-Valued CNNs

CNNs have made great achievements in the field of computer vision, such as LeNet [29], AlexNet [30], Visual Geometry Group (VGG) [31], Residual Network (ResNet) [32], You Only Look Once (YOLO) [33], etc. There is also TextCNN for NLP. CNNs include convolutional layers, pooling layers, and fully connected layers [34]. The convolutional layer is the core of the CNNs. A convolutional layer usually includes multiple convolution kernels of the same size, and the number of convolution kernels determines the size of the output. Similar to classical CNNs, complex-valued convolutional neural networks also contain complex-valued convolutional layers [15]. The complex-valued convolution layer contains the real part convolution and the imaginary part convolution, and the calculation process is shown in Figure 1. $M_R$ and $M_I$ are feature maps with real and imaginary parts, respectively, and $K_R$ and $K_I$ are convolution kernels with real and imaginary parts, respectively. The output of the complex-valued convolutional layer is $M_R K_R - M_I K_I + \mathbf{i}(M_R K_I + M_I K_R)$.

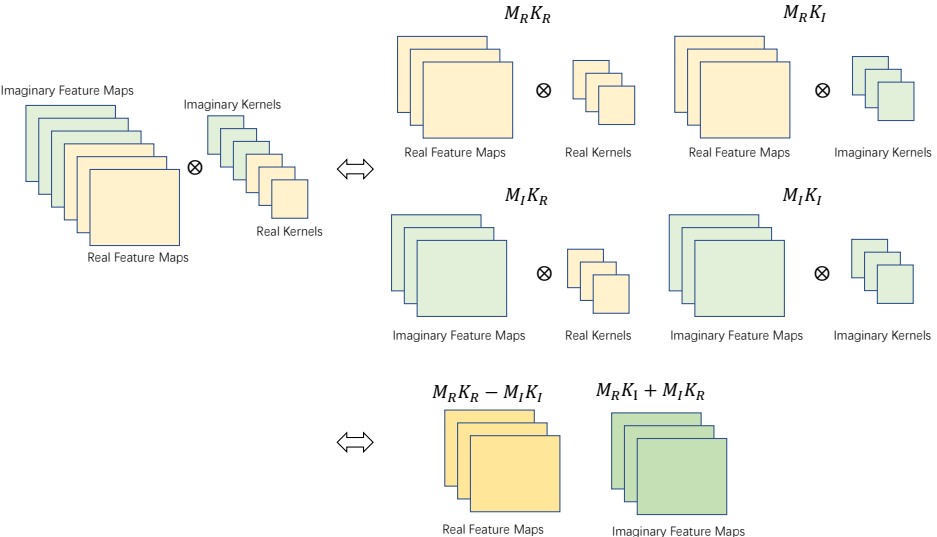

**Figure 1.** Convolutional layer calculation process of complex-valued CNN.

### 2.2.3. Complex-Valued RNN

Unlike linear layers, RNNs are able to learn information from previous neurons because the output of each layer depends on the output of the previous one. RNNs are often used to process sequence data and has a wide range of applications in NLP [35]. LSTM and GRU are two commonly used RNNs. We take complex-valued LSTM as an example to introduce complex-valued RNNs. As shown in Figure 2, LSTM has multiple gates: forget, input, and output gates that selectively let information through. The LSTM is computed as follows [36].

$$f_t = \sigma(W_{xf}X_t + W_{hf}H_{t-1} + b_f), \tag{17}$$

$$i_t = \sigma(W_{xi}X_t + W_{hi}H_{t-1} + b_i), \tag{18}$$

$$o_t = \sigma(W_{xo}X_t + W_{ho}H_{t-1} + b_o), \tag{19}$$

$$\widetilde{C}_t = \tanh(W_{xc}x_t + W_{hc}h_{t-1} + b_c), \tag{20}$$

$$C_t = f_t \cdot C_{t-1} + i_t \cdot \widetilde{C}_t, \tag{21}$$

$$H_t = o_t \cdot \tanh(C_t), \tag{22}$$

where $\cdot$ represents vector dot multiplication, $f_t$ represents forget gate, which is used to discard insignificant information from the past, $i_t$ represents the input gate, $o_t$ represents the output gate, $\widetilde{c}_t$ represents the candidate memory cell, $C_t$ represents the output cell state, and $H_t$ represents the output hidden state.

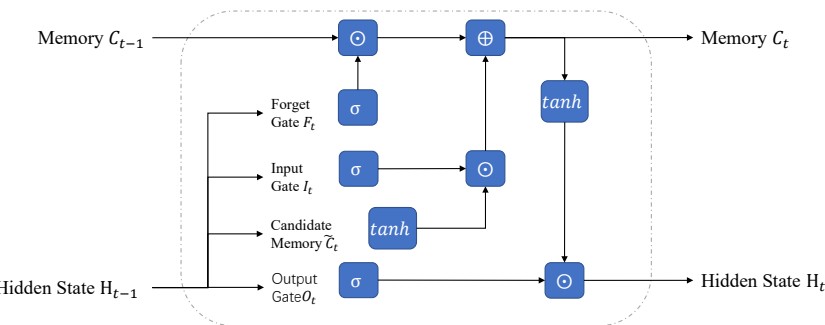

**Figure 2.** The architecture of LSTM.

Complex-valued LSTM requires the forget gate, input gate, and output gate to be implemented using complex-valued linear layers. The sigmoid activation function and tanh function and vector multiplication are also calculated using complex values. Based on complex-valued linear layers, the above procedures are relatively easy to implement.

*2.3. Quantum-Inspired Model*

Sordoni et al. [37] proposed in 2013 the quantum language model, which is the first successful practice of quantum probability theory in NLP. QLM combines quantum theory with the research of NLP, which is of great significance in theory. However, QLM is limited in many ways. For example, the term is represented as a vector in the form of a one-hot vector. Compared with the current commonly used distributed representation, the one-hot vector does not have the ability to take into account the global semantic information of the text, so it takes up more storage space and wastes a lot of storage resources due to its unique representation form. Second, QLM is difficult to embed the density matrix obtained through iterative calculation into an end-to-end neural network that can be trained by a backpropagation algorithm, so it is difficult to obtain practical application. In 2018, Peng Zhang et al. [13] built deep-learning networks using quantum-like mechanics theory based on QLM. The end-to-end quantum-like language models (NNQLM) are proposed to complete the question-answer matching task in NLP [13], which is the earliest combination of quantum-like mechanics theory and deep-learning technology. The application of the end-to-end neural-network model in NLP is realized. However, because the model adopts a real-number vector, it cannot simulate a quantum particle quantum state in a real sense, nor can it make full use of the probability attribute of quantum mechanical density matrix. Therefore, the purpose of the research and implementation of NNQLM is to extend quantum mechanics and NLP theories.

Li et al. [23] adopted a complex word vector to simulate quantum states and proposed quantum-inspired complex word embedding in 2018 . Combined with the theoretical basis of quantum mechanics, they represented text statements in the form of density matrix, used projection measurement to observe and measure text statements in the form of density matrix, and used the measured probability value to infer the polarity of text statements. Then, complete the text classification task for NLP. The density matrix represented by the vector of complex words completely conforms to the theory of quantum mechanics. The network model proposed based on quantum mechanics improves the interpretability of network in NLP tasks. Meanwhile, compared with some classical machine learning models such as Unigram-TFIDF, the model based on complex basis vector simulation quantum state design has better performance on five English binary datasets. Therefore, the model that combines quantum mechanics theory and deep neural network is one of the current research hotspots. Benyou Wang et al. proposed a complex-valued network for matching (CNM) [14] to complete question and answer matching task, which using complex word vectors to simulate quantum states. They used complex density matrix to represent questions-answers, respectively, and carried out projection measurement on the questions and answers in the form of density matrix, respectively. Finally, the similarity of question and answer sentences is evaluated based on the probability value obtained by projection measurement, so as to select the correct answer to the question.

Jiang and Zhang [38] proposed a quantum interference-inspired neural-network matching model (QINM) for processing information extraction tasks in 2020, which could embed interference phenomena into the information extraction process. The experimental results showed that it was superior to the quantum-inspired information extraction models and some neural-network information extraction models previously proposed. Zhang et al. [39] proposed TextTN in 2021, a text tensor network based on quantum theory, for processing text classification tasks. TextTN can be divided into two sub-models. First, word generation tensor network (word-GTN) is used to encode words into vectors, and then sentence discrimination tensor network (sentence-DTN) is used to classify sentences. Zhang et al. [40] proposed a complex-valued fuzzy neural network for conversational

sarcasm recognition, which successfully combines quantum theory with fuzzy logic theory. Shi et al. [41] proposed two end-to-end quantum-inspired deep neural networks ICWE-QNN and CICWE-QNN for text classification. These two models use GRU, CNN and attention mechanism to improve the quantum-inspired model, which can solve the problem of ignoring the internal language characteristics of the text in CE-Mix model.

Quantum-inspired algorithm only uses the mathematical framework of quantum theory and therefore does not need to run on a real quantum computer. Recent applications of the quantum-inspired models in NLP show that this research direction is feasible. Compared with the neural-network model, the advantage of quantum-inspired algorithm is that it can give the model physical meaning, so that the model has better interpretability. In addition, the quantum-inspired algorithm can also embed quantum characteristics such as quantum interference and quantum entanglement into the model, thus enhancing the learning ability of the model.

To sum up, the neural network and quantum mechanics theory can be used together in the field of NLP. At the same time, the existing quantum-inspired models are rarely constructed based on complex-valued neural networks, which do not make full use of the advantages of quantum computing and have gaps in migrating to quantum computers in the future.

## 3. Materials and Methods

In this section, we first introduce the datasets used in our sentiment-classification experiment, then introduce our quantum-inspired fully complex-valued neural network ComplexQNN, and finally introduce the metrics used to evaluate the model and the loss function used in the experiment.

### 3.1. Datasets

We use six sentiment-classification datasets: Customer Review (CR) [42], Opinion polarity dataset (MPQA) [43], Movie Review (MR) [44], Stanford Sentiment Treebank (SST, including SST-2 and SST-5) and Sentence Subjectivity (SUBJ) [44]. The details of the datasets are shown in Table 1 as follows.

**Table 1.** Description of six benchmarking sentiment-classification datasets.

| Dataset | Description | Type | Count |
|:---:|:---:|:---:|:---:|
| CR | Product reviews | pos/neg | 4k |
| MPQA | Opinions | pos/neg | 11k |
| MR | Movie reviews | pos/neg | 11k |
| SST-2 | Movie reviews | pos/neg | 70k |
| SUBJ | Subjectivity | subj/obj | 10k |
| SST-5 | Movie reviews | five labels | 11k |

### 3.2. ComplexQNN

We propose a quantum-inspired fully complex-valued neural network (ComplexQNN) which is also based on the mathematical theory of quantum computing. In the following, we will introduce the theory of the ComplexQNN, the model architecture, the implementation details and the application in sentiment classification.

### 3.2.1. Theory of the ComplexQNN

Quantum-inspired models are used to model natural language in a quantum information way, so the first step is to represent words as quantum states. In a single-atom model, electrons can be in the ground state or the excited state, or in a superposition between the two [20]. Similarly, natural language due to the phenomenon of polysemy can also

be represented as a superposition. A word $w$ has $n$ different semantics ($n = 2^m, m \geq 0$), denoted as $e_i$, then the quantum state of the word is

$$|w\rangle = \sum_{i=1}^{n} \alpha |e_i\rangle, \tag{23}$$

where $\alpha$ is a $n$-dimensional complex-valued vector, and $|\alpha_i|^2$ represents the probability that the word $w$ represents the meaning $e_i$. For example, the quantum state of a word with $n = 4$ is denoted by

$$\begin{aligned} |w\rangle &= \sum_{i=1}^{4} \alpha |e_i\rangle \\ &= \alpha_{00}|e_0\rangle + \alpha_{01}|e_1\rangle + \alpha_{10}|e_2\rangle + \alpha_{11}|e_3\rangle \\ &= \begin{bmatrix} \alpha_{00} \\ \alpha_{01} \\ \alpha_{10} \\ \alpha_{11} \end{bmatrix}. \end{aligned} \tag{24}$$

From the Equation (24), we can see that the quantum state of the word $w$ is mapped to a $n$-dimensional complex vector space. A sentence usually consists of multiple words, just like a quantum system consists of multiple microscopic particles. A quantum system is usually represented by a density matrix in quantum computing. Suppose there are $m$ words in the sentence, then the density matrix of a sentence $S$ is represented as

$$\rho = |S\rangle\langle S| = \sum_{i=1}^{m} \beta |w_i\rangle (\sum_{i=1}^{m} \beta |w_i\rangle)^{\dagger}, \tag{25}$$

where $\beta$ is a $n$-dimensional complex vector, and $|\beta_i|^2$ represents the weight of the word $w_i$ in the sentence $S$. Similar to the attention mechanism, different weights are beneficial for the neural network to focus on the key words in the sentence. In sentiment classification, some adjectives such as "good", "bad", "excellent" have a great influence on the final prediction results, and can be assigned larger weights. After the sentence is represented as a density matrix, we want to further learn the connection between the words in the sentence. Corresponding to the quantum system, this operation is called evolution, i.e., the quantum state changes with time or other external interference, which is represented by

$$\rho' = U\rho, \tag{26}$$

where $U$ is a $n \times n$ complex matrix, and $\rho'$ is the system state after evolution. In the past quantum-inspired language models, it is usually to extract the real part and imaginary part of the density matrix $\rho$, and use RNNs or CNNs to train them separately, and finally integrate the output features. We believe that this operation will cut off the information in the quantum system, which will lead to incomplete features learned and cannot correctly simulate the change of the quantum system state. Therefore, when constructing the quantum-inspired model, we simulate the change of the quantum state through complex-valued neural networks, and the whole evolution process will be based on complex values, and the output result will also remain in the complex state.

Finally, the measurement in quantum computing can obtain the probability value of the quantum system collapsing to a set of base states, which is applied to the text classification task in natural language processing. Suppose $M_i(i = 1, \ldots, k)$ is a set of measurement operators representing $k$ classification labels. The measurement probability of the sentence corresponding to the $i$-th label is

$$p_i = \rho^{\dagger} M_i^{\dagger} M_i \rho. \tag{27}$$

We applied the model to the sentiment analysis task to verify the sentiment polarity of sentences, including binary classification and multi-classification, and the details of the experiment can be seen in Section 4. The following introduce the ComplexQNN from three aspects: the architecture of the ComplexQNN, the implementation details of the ComplexQNN, and its application in sentiment analysis.

### 3.2.2. Architecture of the ComplexQNN

The architecture of the ComplexQNN is depicted in Figure 3. We can see that it consists of four modules: complex embedding, projection, evolution and classifier. First, the input data of the ComplexQNN need to obtain through preprocessing, as with case conversion, word segmentation, word index mapping, filling and truncation. Moreover, in order to mask the additional token sequence brought by filling sequence, the mask sequences composed of 0 and 1 is also needed to be constructed. To sum up, The token sequences and mask sequences are the input data of the ComplexQNN. The following describes four essential modules of the ComplexQNN.

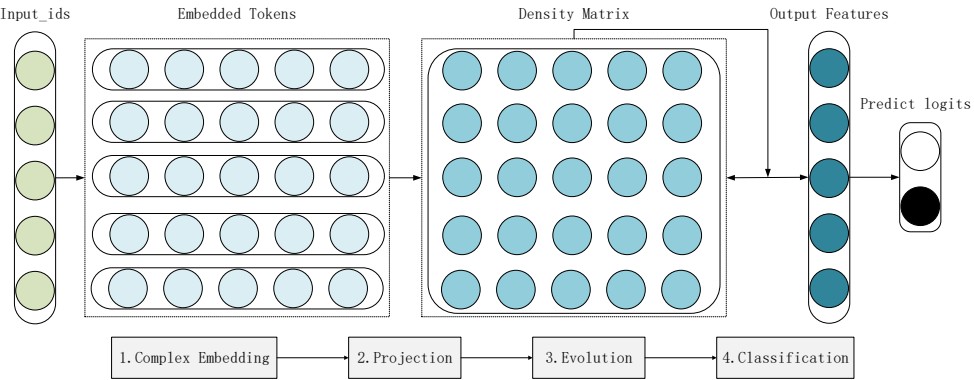

**Figure 3.** The architecture of the ComplexQNN.

1. Complex embedding: The complex embedding is to map the token number corresponding to the word (which is the position of the corresponding integer word in the vocabulary) into the n-dimensional complex vector space, which corresponds to the quantum state construction process in quantum computing. Each word is mapped from discrete space to high-dimensional Hilbert space, which corresponds to a complex-valued column vector.

2. Projection: The projection is the mapping of discrete words in a sentence into a complex value space of $n \times n$. In the previous step, plural word embeddings have mapped words into an $n-$dimensional complex-valued vector space. From Equation (25), the density matrix representation of the sentence can be calculated, where the weight $\beta$ of the words can be trained by the attention mechanism, and by default all words take the same weight.

3. Evolution: The evolution process is to simulate the change of the quantum system. In the theory of quantum computation, the change of quantum state and density matrix is realized using quantum gates. A quantum gate corresponds to a unitary matrix whose dimension corresponds to the number of qubits it operates on ($n = 2^m$). In the ComplexQNN, we simulate the changes of quantum systems through complex-valued linear layers, complex-valued recurrent neural networks and complex-valued convolutional neural networks. The dimensions of the input and output of the evolution module that we designed are both $n \times n$. Therefore, the dimension of the original quantum system will not be changed after learning the features inside the sentence.

4. Classifier: The classifier uses the high-dimensional features learned by the previous modules as input to predict the classification result. Based on the theory of quantum computing, we can directly predict the output from the measurement, as shown in

Equation (27). It is necessary to construct a set of linearly independent measurement bases. The number of measurement bases depends on the number of classification results performed. The result predicted by the final model takes the label corresponding to the measurement basis with the largest probability value.

The above introduces the four essential modules of the ComplexQNN and shows the progression of text sequences from input to predictive output. The specific design of the module is described as follows.

### 3.2.3. Implementation Details of the ComplexQNN

The ComplexQNN needs to include complex word embedding, projection, evolution, and classifier among which complex word embedding and evolution are the core of model construction. In our implementation, we designed three modules using Allennlp (https://allenai.org/allennlp, accessed on 25 January 2018) library: complex embedder, quantum encoder and Classifier. Projection and evolution operations are included in the quantum coding layer.

The complex embedder is the first module of the ComplexQNN. Its input is the preprocessed text Token sequence, which is an integer vector. Complex word embedding consists of real and imaginary embedding layers. Figure 4 shows the processing of complex word embedding. The text Token sequence is passed through these two embedding layers, respectively, and finally the complex word vector representation of each Token is calculated by Equation (28). The real and imaginary embedding layers can conveniently use classical word embedding as

$$[w_1, \ldots, w_i, \ldots, w_n] = [r_1, \ldots, r_i, \ldots, r_n] + \mathbf{i} \times [i_1, \ldots, i_i, \ldots, i_n]. \tag{28}$$

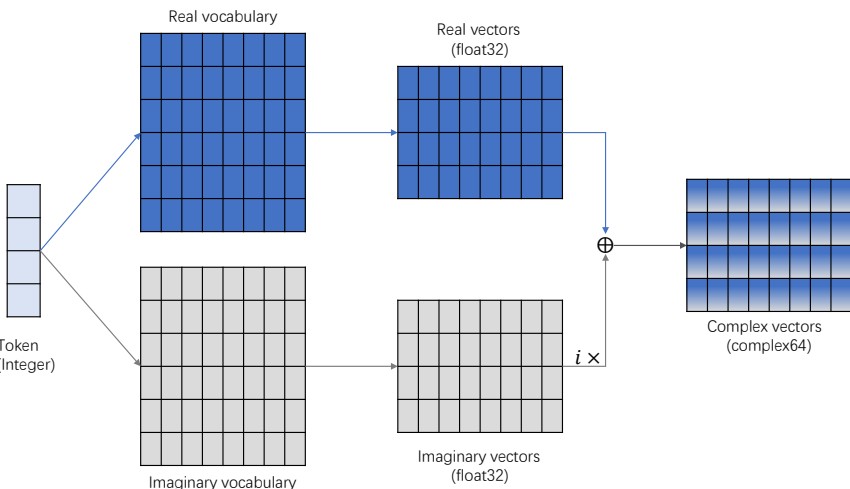

**Figure 4.** The processing of complex word embedding.

The classic word embedding layer has many different embedding methods. According to the level of word segmentation, it can be divided into character level, wordpiece level and word level. According to the training method, it can be divided into non-contextual word embedding (static word embedding) and contextual word embedding (dynamic word embedding) [45]. Word2vec [46] and GloVe [47] are two classic non-contextual word embedding methods. Context word embeddings are derived from the pretraining models, and the contextual word vectors related to the downstreaming dataset is obtained through fine-tuning.

We have considered different schemes to construct the complex word embedding layer. (1) Use different pretraining models as the embedding layer of the real and imaginary parts, such as BERT for the real part and RoBERTa for the imaginary part, so that different information can be encoded into the quantum state. (2) The real part and the imaginary part

encode different types of text information. For example, the real part encodes wordpiece-level word vectors, and the imaginary part uses the NLTK library to encode semantic information such as word polarity. (3) The real part encodes the forward word order feature, and the imaginary part encodes the reverse word order feature.

The first method, which uses different pretraining models to extract features at the same time, can achieve the best experimental results, but it requires a lot of memory resources for training. The second and third methods are fast to train, but the results are not as good as the pretraining models. Considering the experimental results and the resources required for training, RoBERTa is used as the real part of ComplexEmbedder, and self-training word embedding layer is used as the imaginary part. In general, we hope that the real part and the imaginary part embed different types of text features, make full use of the heterogeneous characteristics of complex-valued neural networks, and then improve the semantic expression ability of the model.

The second module of the ComplexQNN is a quantum encoder for projection and evolution. As we mentioned earlier, we need to projection obtaining the density matrix representation of the sentence. The Equation (25) shows the process of calculating the density matrix. Evolution is the process of simulating the operation of quantum gates. This needs to meet some conditions, i.e., the input and output dimensions are unchanged and the dimension is $2^n$. We build our encoding layers based on the following basic building blocks: complex-valued fully connected layers, complex-valued recurrent neural networks, and complex-valued convolutional neural networks [15]. We construct three intermediate module layers for encoding: a complex-valued deep neural-network encoding layer, a complex-valued recurrent neural-network encoding layer (based on ComplexLSTM), and a complex-valued convolutional neural-network encoding layer.

As shown in Figure 5, we construct the ComplexTextCNN. The input of this module is the projected density matrix $\rho$ representing the sentence. We use three different convolution kernel sizes with $[3, 4, 5]$, the number of each convolution kernel is 2, and the stride is 1. Then, the features are extracted by max pooling, and the features learned in different dimensions are concated together. So far, the biggest difference between the ComplexTextCNN and classic TextCNN is that all calculation operations are calculated in complex-valued networks. Finally, in order to obtain a new sentence density matrix representation after learning features, we use a complex-valued fully connected network layer to restore the vector dimension to the input dimension. Through the outer product operation, the matrix form of the same dimension as the input is obtained.

The third module is a classifier, which predicts the output of the model based on quantum-computing measurements. Specifically, it is realized through Equation (27). We can design different numbers of measured ground states for text multi-classification. In the following experiments, the number of measurement basis vectors needs to be determined according to the number of classification labels. The final classification result is to take the one with the largest predicted value as the final result. In addition, considering that the output of the measurement is a real value, we can take the probability value of each label to construct a prediction vector, and then splice it with the prediction results of other models to achieve model fusion and achieve better experimental results.

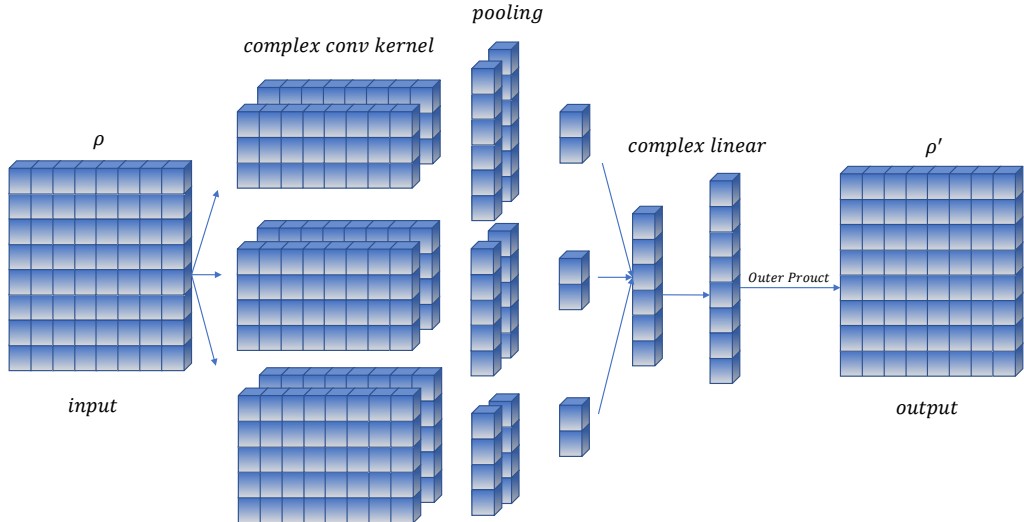

**Figure 5.** The architecture of ComplexTextCNN.

### 3.2.4. Application in Sentiment Analysis

Sentiment classification is a common task in NLP, which aims to predict the sentiment polarity corresponding to a sentence. We conduct this task to validate the experimental performance of the ComplexQNN. At the same time, we will compare some classic network models. Figure 6 is the flow chart of the ComplexQNN for sentiment-classification task, and the data dimensions are annotated. First, the text is preprocessed to normalize case, segment words and remove stop words. Second, the quantum states of words are simulated by the complex word embedding layer. The quantum encoder layer is then used for projection and evolution. Finally, the prediction results are output by the simulated measurement operation of the classifier.

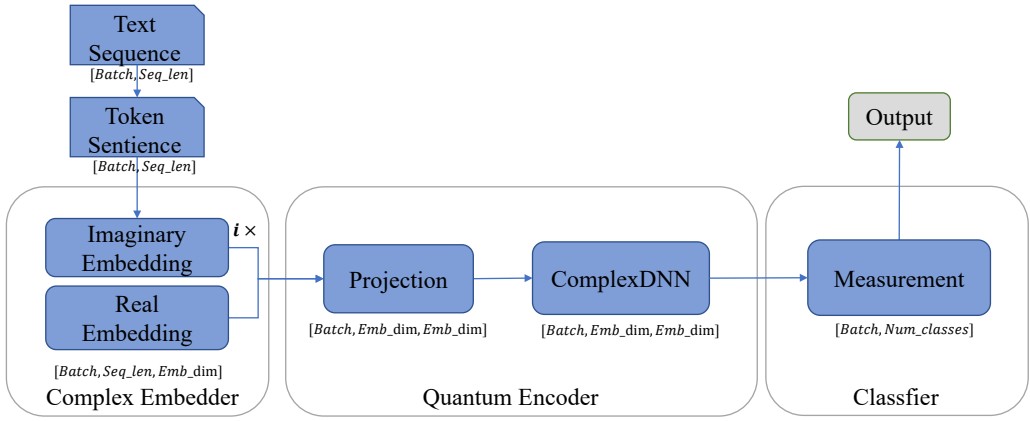

**Figure 6.** The flow chart of the ComplexQNN for sentiment-classification tasks.

### 3.3. Evaluation Metrics

We use five sentiment binary classification datasets (CR, MPQA, MR, SST-2 and SUBJ) and a sentiment five classification dataset (SST-5). In the sentiment-classification experiment, we use two evaluation metrics (accuracy and F1-score) to comprehensively evaluate the performance of different models. The specific calculation methods are as follows.

- Accuracy: Accuracy is the ratio of correctly predicted terms to the total terms:

$$\text{Accuracy} = \frac{TP + TN}{TP + TN + FP + FN}. \tag{29}$$

TP, TN, FP and FN denote the true positives, true negatives, false positives and false negatives, respectively.
- F1-score: F1-score is the harmonic mean of Precision and Recall:

$$\text{F1-score} = 2 \times \frac{Precision \times Recall}{Precision + Recall}. \tag{30}$$

The calculation of precision and recall is as follows:

$$\text{Precision} = \frac{TP}{TP + FP}, \ \text{Recall} = \frac{TP}{TP + FN}. \tag{31}$$

The loss function we used in our experiment is the cross-entropy loss defined as

$$Loss = -\frac{1}{n}\sum_i [y_i \log(p_i) + (1 - y_i)\log(1 - p_i)], \tag{32}$$

where $n$ is the number of samples, $y_i$ is the $i$-th label and $p_i$ is the predicted probability value of positive class.

## 4. Experimental Results and Discussion

In this section, we conduct experiments on sentiment classification to verify our proposed method (Our code is available at Github: https://github.com/Levyya/ComplexQNN, accessed on 15 March 2023). We introduce the models used for comparison, present and analyze our experimental results, and discuss the advantages of ComplexQNN compared to classical neural networks.

### 4.1. Comparison Models

We use five classical models, TextCNN, GRU, Embeddings from Language Models (ELMo), Bidirectional Encoder Representations from Transformers (BERT) and Robustly Optimized BERT Pretraining Approach (RoBERTa), as the comparison models for our experiments, which are described below.

TextCNN [48] is built on convolutional neural networks with pretrained vectors word2vec. The experimental results shown below are CNN-non-static model .

GRU is a kind of RNNs. Like LSTM, it is also designed to solve long-term memory and gradient problems, but GRU is faster than LSTM.

ELMo [49] is composed of bidirectional LSTMS as the basic components. With language model as the training target, ELMo is pretrained with large corpus to obtain a common semantic representation, which is then migrated to the downstream NLP task. It can significantly improve the model performance of downstream tasks. ELMo provides word-level semantic representation and performs well in many downstream tasks.

BERT [6] is a pretraining language model (PLM). For example, ELMo and GPT are Auto Regressive (AR) models, which only consider unilateral information, i.e., predicting the next word with reference to previous words with reference to context. BERT reconstructs original data from noisy data using context information, which belongs to Auto Encoding model. Two tasks were used during pretraining: Masked Language Model (MLM) and Next Sentence Prediction (NSP). The output of BERT is the 768-dimensional vector for each Token in the sentence, and a special Token ([CLS]).

RoBERTa [7] is an improved version of BERT that uses larger model parameters, larger batch sizes, and more pretraining data, while improving the training method and removing the NSP task. Using dynamic mask and BPE Encoding (Byte-Pair Encoding), the experimental results are better than BERT's.

*4.2. Results and Analysis*

In the experiment of sentiment classification, we choose some classic models in NLP such as TextCNN, GRU, and popular pretraining models such as ELMo, BERT and RoBERTa as comparison models. Tables 2 and 3 show the experimental results on six sentiment-classification datasets. The former is the experimental results under the accuracy metric, while the latter is the experimental result under the F1-score metric. Figure 7 shows the graph of experimental results under F1-score. The following is a detailed analysis of the experimental results.

**Table 2.** Experimental results on six benchmarking sentiment-classification datasets evaluated with accuracy.

| Model | CR | MPQA | MR | SST-2 | SUBJ | SST-5 |
|---|---|---|---|---|---|---|
| TextCNN | 78.8 | 74.4 | 75 | 81.5 | 90.3 | 34.9 |
| GRU | 80.1 | 84.3 | 76 | 81.6 | 91.7 | 37.6 |
| ELMo | 85.4 | 84.4 | 81 | 89.3 | 94.9 | 48 |
| BERT | 88.8 | 89.5 | 84.9 | 93 | 95.2 | 52.5 |
| RoBERTa | 90.4 | 90.9 | 89.8 | 88.8 | 96.7 | 53.1 |
| ComplexQNN | 91.2 | 91.5 | 89.9 | 90.4 | 97.3 | 54.3 |

**Table 3.** Experimental results on six benchmarking sentiment-classification datasets evaluated with F1-score.

| Model | CR | MPQA | MR | SST-2 | SUBJ | SST-5 |
|---|---|---|---|---|---|---|
| TextCNN | 71.2 | 75.5 | 75.9 | 80.9 | 90.1 | 48.8 |
| GRU | 71 | 73.3 | 75.5 | 82.3 | 91.8 | 46.7 |
| ELMo | 79.9 | 74.2 | 82.3 | 88.4 | 94.9 | 48.3 |
| BERT | 85.2 | 82.3 | 84.6 | 88 | 95.2 | 52.5 |
| RoBERTa | 85.8 | 85.6 | 89.9 | 89.7 | 96.7 | 52.7 |
| ComplexQNN | 86 | 86.4 | 90.3 | 88.4 | 97.3 | 53.1 |

1.  Compared with classical models TextCNN and GRU, popular pretraining models (ELMo, BERT, RoBERTa) and our model has significant advantages. Specifically, we show the improvement effect of the ComplexQNN by comparing the results. The comparison results of the ComplexQNN and TextCNN under the accuracy metric for the considered datasets are: CR (+12.4%), MPQA (+17.1%), MR (+14.9%), SST-2 (+8.9%), SUBJ (+7.0%), SST-5 (+19.4%), average improvement (+13.28%); Comparison results of the ComplexQNN and TextCNN under the F1-score metric: CR (+14.8), MPQA (+10.9), MR (+14.4), SST-2 (+7.5), SUBJ (+7.2), SST-5 (+4.3), average improvement (+9.85). The above data show that the ComplexQNN has a great performance improvement compared to TextCNN, because the ComplexQNN is a network model designed based on quantum-computing theory, which can learn more complex text features.

2.  Compared with the pretraining models, the ComplexQNN has better experimental results than ELMo and BERT. Compared to RoBERTa, ComplexQNN has better results in CR, MPQA, MR, SUBJ and SST-5 (under the F1-score metric): CR (+0.2), MPQA (+0.8), MR (+0.4), SUBJ (+0.6), SST-5 (+0.4); on the SST-2 dataset, the F1-score result of the ComplexQNN is slightly lower than that of RoBERTa (−1.3), but the ComplexQNN has a better experimental result (+1.6%) under the accuracy metric. From a numerical point of view, the improvement of ComplexQNN compared to RoBERTa is small, but in fact this is because RoBERTa has achieved good experimental results on these six datasets, which is why our model is slightly lower than RoBERTa on one of these datasets.

3.  From Figure 7, we can clearly see that the ComplexQNN is almost always at the highest level (except for slightly lower results in the SST-2 dataset), which indicates

that the ComplexQNN has a significant performance advantage over the six sentiment-classification datasets.

4.  Table 4 shows the training time of six different models on six sentiment-classification datasets. We use an Nvidia 2080Ti GPU and set the batch size as 32. The results are use the format of "minutes:seconds", and it means that a model costs training time on an epoch. According to this table, we can see that in order of training speed from fast to slow, they are TextCNN, GRU, ELMo, BERT (RoBERTa), and ComplexQNN. The reason about our model need the most training time is that the ComplexQNN uses classical computation to simulate quantum computation. Complex-valued calculations require additional imaginary part parameters to simulate. However, we can see that our model can also complete the training in a very short time.

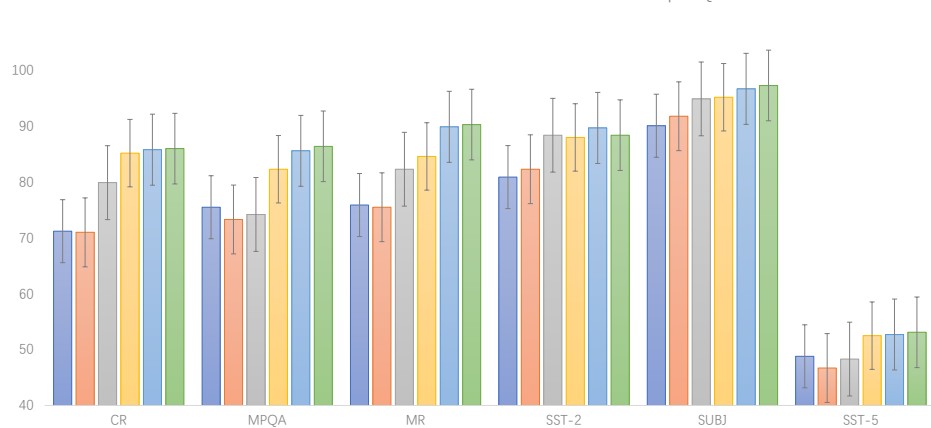

**Figure 7.** The F1-score results of six models on six sentiment-classification datasets.

**Table 4.** The training time of six models on six sentiment-classification datasets.

| Model | Training Time | | | | | |
|---|---|---|---|---|---|---|
| | **CR** | **MPQA** | **MR** | **SST-2** | **SUBJ** | **SST-5** |
| TextCNN | 0:03 | 0:07 | 0:07 | 0:47 | 0:04 | 0:05 |
| GRU | 0:04 | 0:04 | 0:10 | 1:25 | 0:09 | 0:09 |
| ELMo | 0:12 | 0:12 | 0:30 | 2:39 | 0:33 | 0:53 |
| BERT | 0:12 | 0:15 | 0:33 | 4:31 | 0:38 | 0:35 |
| RoBERTa | 0:12 | 0:15 | 0:33 | 4:31 | 0:38 | 0:35 |
| ComplexQNN | 0:32 | 0:32 | 1:17 | 10:23 | 1:48 | 1:34 |

### 4.3. Discussion

ChatGPT (https://openai.com/api/, accessed on 30 November 2022) has been a big shock, and it has made many people realize that NLP technology based on classical neural networks has reached a very high level. RoBERTa is an excellent language model in the NLP area, achieving human-level performance on many datasets. Therefore, we are not surprised that the results of our proposed model ComplexQNN are close to RoBERTa on multiple datasets. Below, we discuss some of the advantages ComplexQNN has over popular classical models such as RoBERTa. We summarize the following three points.

First, let us look at the difference between ComplexQNN and RoBERTa. RoBERTa is an improvement on the BERT model. It is a multi-layer transformer structure, which can extract features in real value space. The ComplexQNN is defined in the Hilbert space, the word vector will be represented as a complex value, and the representation space is larger than the real value space. The real part can represent the semantic information of the context, and the imaginary part can be used to represent the information outside the semantics, such as the position information of the words in the sentence, the sentiment

information of the words, and the ambiguity information of the words. Compared with the real-number space, the complex number space gives deep-learning algorithms more representation possibilities, which is conducive to expanding the boundaries of model development.

Second, most classical models are black-box models. In the existing natural-language model, the text is mapped to a vector, and then through the multi-layer neural-network structure, and the meaning of the intermediate vector can only be described by the low-dimensional and high-dimensional features of the text. The quantum-inspired model based on quantum computing regards natural language as a quantum system: words are represented as quantum states, sentences are represented as density matrices, interactions of words in sentences are represented as quantum state evolution, and the corresponding labels of sentences are represented as quantum state measurement followed by collapse to the ground state. This brings physical meaning to the model, which is beneficial to people's understanding of the model. Some characteristics of natural language can be explained by quantum phenomena, i.e., polysemy of words can be well represented by quantum entanglement, which increases the interpretability of the model to a certain extent.

Finally, there is the issue of computational complexity. As of now, it takes twice as many resources (real and imaginary parts) to implement a quantum-inspired complex-valued network, but that is because of simulating quantum operations in a classical computer. Since $n$ qubits need to be simulated with $2^n$ classical bits (or $2^{n+1}$ when considering complex values), an $n-$bits quantum gate needs $2^n \times 2^n$ classical bits to be simulated. The neural-network layers designed by ComplexQNN are based on complex-valued neural networks, which are easily transferable to future quantum computers. However, when our algorithm runs in a real quantum computer, the storage and computing resources spent will be exponentially reduced. Existing quantum computers have exceeded 100 qubits and are capable of handling classification tasks on small-scale natural-language-processing datasets. We look forward to implementing our proposed algorithms in real quantum computers in the future.

In summary, compared to classical neural-network models such as RoBERTa, ComplexQNN has stronger representation power, better interpretability, and the possibility of exponential complexity reduction. However, we still face some challenges to overcome. First, this algorithm uses classical neural networks, including complex-valued neural networks, to simulate the quantum-computing process. Therefore, additional parameters are required to represent the imaginary parts. Second, the complex-valued network space is more complex than the real-number space, and the design of complex-valued networks, such as optimizers and network modules, needs further research. Third, the scale of existing quantum computers is too small and too expensive to use. It is difficult to completely migrate the existing quantum-inspired methods to real quantum computers.

## 5. Conclusions

We propose a quantum-inspired fully complex-valued model, ComplexQNN, based on complex-valued neural networks. We describe the construction principle of the model as well as the implementation details and verify the effectiveness of ComplexQNN on six sentiment-classification datasets, including binary and multi-classification. Future research could consider two directions: the first is to encode deeper semantics, such as by developing complex-valued transformer network modules suitable for larger datasets and applying them in more complex scenarios, such as machine translation and recommendation systems; the second is to build network modules using quantum-circuit models, although this is limited in the dataset that can be processed in the NISQ era.

**Author Contributions:** Conceptualization, W.L., J.S. and Y.C.; methodology, W.L., J.S. and Y.C.; software, W.L. and J.S.; validation, W.L. and J.S.; formal analysis, W.L. and J.S.; investigation, W.L., J.S. and Y.C.; resources, W.L., J.S. and Y.C.; data curation, W.L., J.S. and Y.C.; writing—original draft preparation, W.L.; writing—review and editing, W.L., J.S. and Y.C.; visualization, W.L.; supervision,

J.S.; project administration, W.L., J.S. and Y.C.; funding acquisition, J.S. All authors have read and agreed to the published version of the manuscript.

**Funding:** This research was funded by the National Natural Science Foundation of China (Grant Nos. 62272483, 61972418), the Education Department of Hunan Province of China (Grant Nos. 22B0001), the Special Foundation for Distinguished Young Scientists of Changsha (Grant Nos. kq1905058) and the Open Fund of Advanced Cryptography and System Security Key Laboratory of Sichuan Province (Grant Nos. SKLACSS-202107).

**Data Availability Statement:** We have used six datasets including: Customer Review (CR) [42], Opinion polarity dataset (MPQA) [43], Movie Review (MR) [44], Stanford Sentiment Treebank (SST, including SST-2 and SST-5) and Sentence Subjectivity (SUBJ) [44].

**Conflicts of Interest:** The authors declare no conflict of interest.

## Abbreviations

The following abbreviations are used in this manuscript:

| | |
|---|---|
| CNN | Convolutional Neural Network |
| RNN | Recurrent Neural Network |
| LM | Language Model |
| QLM | Quantum Language Model |
| QNN | Quantum Neural Network |
| NISQ | Noisy Intermediate-Scale Quantum |
| LSTM | Long Short-Term Memory |
| GRU | Gated Recurrent Unit |
| ELMo | Embeddings from Language Models |
| BERT | Bidirectional Encoder Representations from Transformers |
| RoBERTa | Robustly Optimized BERT Pretraining Approach |
| CR | Customer Review |
| MPQA | Multi-Perspective Question Answering |
| MR | Movie Review |
| SST | Stanford Sentiment Treebank |
| SUBJ | Sentence Subjectivity |
| VGG | Visual Geometry Group |
| ResNet | Residual Network |
| YOLO | You Only Look Once |
| CTQW | Continuous-Time Quantum Walk |
| CNM | Complex-valued Network for Matching |
| TextTN | Text Tensor Network |
| NNQLM | End-to-End Quantum-like Language Models |
| GTN | Generation Tensor Network |
| DTN | Discrimination Tensor Network |
| ICWE | Interpretable Complex-valued Word Embedding |
| CICWE | Convolutional Complex-valued Neural Network based on ICWE |
| ComplexQNN | Quantum-inspired Fully Complex-Valued Neural Network |

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
