# Peer review of "Quantum-Inspired Fully Complex-Valued Neutral Network for Sentiment Analysis"

_axioms, doi:10.3390/axioms12030308_

Round 1

Reviewer 1 Report

Dear authors,

Thank you for submitting your paper. I enjoyed reading it, but I do have some concerns which I have highlighted in the attached file. Additionally, there are some grammatical mistakes that I would like you to address. Please carefully review the paper and make the necessary corrections.

Thank you for your attention to this matter.

Best regards,

Author Response

Dear Reviewer

In regards to the following

Article Number: axioms-2260415

“Quantum-Inspired Fully Complex-Valued Neutral Network for Sentiment Analysis”

Thank you very much for your instructive and constructive comments.

Those comments are all valuable and very helpful for revising and improving our paper, and with the important guiding significance to our researches. We have tried our best to revise and improve the manuscript according to the reviewers’ thoughtful and constructive comments. The manuscript has now been duly revised by taking into account all comments made by both editors and reviewers, and special changes are listed on the subsequent pages of this letter. Electronic versions of both this letter and the revised manuscript are included as well.

We appreciate for editors’ and reviewers’ work earnestly, and hope that the corrections will meet with approval. Once again, thank you very much for your comments and suggestions. We look forward to your information about our revised manuscript.

In closing, we would like to thank you for the much time and effort in handling our manuscript and in providing helpful improvement suggestions.

With best regards,

Yours sincerely,

Wei Lai

Jin-jing Shi

Yan Chang

School of Computer Science and Engineering,

Central South University,

Changsha, 410083, China

Response to Reviewer 1 Comments

We would like to thank you for a lot of very constructive and valuable comments which have great guiding significance for our researches.

Point 1: Is this equation a special case of Schrödinger equation? if so, please mention it.

Are you sure that this equation is correct? In my opinion, it is not correct. Please explain how you can obtain it?

Response 2: The quantum state is the basic concept in quantum computation theory, which is often noted with Dirac symbol, likes  and . These two equation are from Nielsen et al.’s book “Quantum Computation and Quantum Information”. The book is available at http://csis.pace.edu/~ctappert/cs837-19spring/QC-textbook.pdf. The first equation can be seen on Page 13 with Eq. (1.1). And the second can be seen on Page 15 with Eq. (1.4).

Point 2: There are many abbreviations in the paper that they are not in this list.

Response 2: Thanks for the reviewer’s constructive and valuable suggestion for the improvement of this manuscript. We have carefully checked the shortcuts word used in our manuscript and revised those incorrect words.

Detailed revisions as follows:

  1. We have revised the “NISQ” to “noisy intermediate-scale quantum (NISQ)” on Line 30.
  2. We have revised the “VGG” to “Visual Geometry Group (VGG)” on Line 248.
  3. We have revised the “ResNet” to “Residual Network (ResNet)” on Line 248.
  4. We have revised the “Yolo” to “You Only Look Once (YOLO)” on Line 249.
  5. We have updated the abbreviations used in our manuscript on Line 687, including ELMo, BERT, RoBERTa, CR, MPQA, MR, SST, SUBJ, VGG, ResNet, YOLO, CTQW, CNM, TextTN, NNQLM, GTN, DTN, ICWE, CICWE, and ComplexQNN.

Point 3: Additionally, there are some grammatical mistakes that I would like you to address.

Response 3: We thank the reviewer’s earnest and sincere comment. We feel sorry for our carelessness. The mistakes have been corrected in the revised manuscript.

Detailed revisions as follows:

  1. The expressions “Therefore, we propose a quantum-inspired fully complex-valued model, ComplexQNN. Our model builds a complex-valued embedding layer, \added{a} quantum encoding layer, and a measurement layer completely based on \added{a} complex-valued neural network.” have been revised into “Therefore, we propose a new quantum-inspired model for NLP, ComplexQNN, which contains a complex-valued embedding layer, a quantum encoding layer, and a measurement layer. The modules of ComplexQNN are fully based on complex-valued neural networks.”.
  2. We have revised the sentence “We conduct experiments on six sentiment classification datasets\added{,} and ComplexQNN can achieve competitive results compared with some classical models such as ELMo, BERT, and RoBERTa” to “We conduct experiment on six sentiment classification datasets comparing with five classical model (TextCNN, GRU, ELMo, BERT, and RoBERTa). The results show that our model has improved by 10% in accuracy metric compared with TextCNN and GRU and has competitive experimental results with ELMo, BERT, and RoBERTa.”.
  3. The word “map” on Line 2 has been revised into “maps”. The word “model” on Line 6 has been revised into “modeling”. The word “under” on Line 9 has been revised into “in”. The words “be transferred” have been revised to “transfer” on Line 16. We have revised the situation of “in the future” on Line 17.
  4. The expression “However, the problem of gradient explosion and gradient disappearance of RNNs make it difficult to build deep neural network.” has been revised into “But gradient explosion and gradient disappearance are problems with RNNs that make it hard to build a deep neural network.” on Line 47.
  5. The expression “These pre-training models can be used as text encoders, which can be easily adapted to different downstream tasks by fine-tuning strategies.” has been revised into “By using fine-tuning techniques, these pre-training models can serve as text encoders that are easily adaptable to various downstream tasks.” on Line 54.
  6. The expression “It has been proved that in some tasks, quantum computers have exponential computational complexity acceleration advantage.” has been revised into “It has been demonstrated that quantum computers have an exponential computational complexity acceleration advantage for some tasks.” on Line 63.
  7. The expression “Interestingly, some researchers build quantum-inspired models based on the mathematical framework of quantum computing theory.” has been revised into “Some researchers use the math behind quantum computing theory to build models that look like quantum computers.” on Line 82.
  8. The expression “A deep complex-valued convolutional network is proposed by Chiheb Trabelsi et al. [15] and has achieved” has been revised into “Trabelsi et al. [15] proposed a deep complex-valued convolutional network that has demonstrated” on Line 96.
  9. We feel sorry that we can not list all the revisions. More revisions can be seen in the revised manuscript. We hope you can understand. Thank you for your sincere comments again.

In brief, we have tried our best to revise and improve the manuscript and made great changes in the manuscript according to the editors’ and reviewers’ constructive and thoughtful comments, and we hope that the corrections will meet with approval. Once again, we would like to thank all editors and reviewers very much for their valuable comments and suggestions that greatly help to improve the presentation of the paper.

Reviewer 2 Report

This paper presents a novel method with an attractive application, which probably can be published and gain future research attention. Therefore, some advices can ne introduced to improve the paper quality as follows.              

1.      The abstract can be improved to present the research main idea for those unfamiliar with this domain easily. Also, the details of the conducted contributions, experiments, comparisons, and obtained results should be added into the abstract section properly.

2.      The introduction section can be presented in another way. The authors can give the introduction section in some terms like the general idea that already demonstrates an interest these days, the particular domain that uses the current application, the main problem in this research, and the central gap founded by the authors in this paper. Some related works support the main claim in this paper and support this work by focusing on some issues. At the end of the introduction section, the authors should be given a clear and comprehensive paragraph to show the readers how this research has been done (the main problem, contributions, experiments, comparisons, results, and so on).

3.      Some shortcuts word need to be justified like NISQ

4.      A few of the references in Related Work are old. Authors need to refer to recent publications in this area.

5.      The authors should give the readers a straightforward method of choosing the experiments and their design. This will help the future researcher in this domain conduct new research and start from the current paper.

6.      The discussion of the results needs to include the strengths and weaknesses of the proposed algorithm.

Author Response

Dear Reviewer

In regards to the following

Article Number: axioms-2260415

“Quantum-Inspired Fully Complex-Valued Neutral Network for Sentiment Analysis”

Thank you very much for your instructive and constructive comments.

Those comments are all valuable and very helpful for revising and improving our paper, and with the important guiding significance to our researches. We have tried our best to revise and improve the manuscript according to the reviewers’ thoughtful and constructive comments. The manuscript has now been duly revised by taking into account all comments made by both editors and reviewers, and special changes are listed on the subsequent pages of this letter. Electronic versions of both this letter and the revised manuscript are included as well.

We appreciate for editors’ and reviewers’ work earnestly, and hope that the corrections will meet with approval. Once again, thank you very much for your comments and suggestions. We look forward to your information about our revised manuscript.

In closing, we would like to thank you for the much time and effort in handling our manuscript and in providing helpful improvement suggestions.

With best regards,

Yours sincerely,

Wei Lai

Jin-jing Shi

Yan Chang

School of Computer Science and Engineering,

Central South University,

Changsha, 410083, China

Response to Reviewer 2 Comments

We would like to thank the reviewer for very constructive and valuable comments which have great guiding significance for our researches.

Point 0: This paper presents a novel method with an attractive application, which probably can be published and gain future research attention. Therefore, some advices can be introduced to improve the paper quality as follows.              

Point 1: The abstract can be improved to present the research main idea for those unfamiliar with this domain easily. Also, the details of the conducted contributions, experiments, comparisons, and obtained results should be added into the abstract section properly.

Response 1: We thank the reviewer’s earnest and sincere comment. Our reseach main idea is to propose a new quantum-inspired model with complex-valued neural network named ComplexQNN. The contributions are including that we build a complex-valued embedding layer, a quantum encoding layer, and a measurement layer. And we conduct experiment on six sentiment classification datasets comparing with five classical model (TextCNN, GRU, ELMo, BERT, and RoBERTa). The results show that our model has improved by 10% compared with TextCNN and GRU and has competitive experimental results with ELMo, BERT, and RoBERTa. We have made the following revisions to highlight our main idea and conducted contributions.

Detailed revisions as follows:

  1. The expressions “Therefore, we propose a quantum-inspired fully complex-valued model, ComplexQNN. Our model builds a complex-valued embedding layer, \added{a} quantum encoding layer, and a measurement layer completely based on \added{a} complex-valued neural network.” have been revised into “Therefore, we propose a new quantum-inspired model for NLP, ComplexQNN, which contains a complex-valued embedding layer, a quantum encoding layer, and a measurement layer. The modules of ComplexQNN are fully based on complex-valued neural networks.” on Line 10.
  2. We have revised the sentence “We conduct experiments on six sentiment classification datasets\added{,} and ComplexQNN can achieve competitive results compared with some classical models such as ELMo, BERT, and RoBERTa” to “We conduct experiment on six sentiment classification datasets comparing with five classical model (TextCNN, GRU, ELMo, BERT, and RoBERTa). The results show that our model has improved by 10% in accuracy metric compared with TextCNN and GRU and has competitive experimental results with ELMo, BERT, and RoBERTa.” on Line 17.
  3. The word “map” on Line 2 has been revised into “maps”. The word “model” on Line 6 has been revised into “modeling”. The word “under” on Line 9 has been revised into “in”. The words “be transferred” have been revised to “transfer” on Line 16. We have revised the situation of “in the future” on Line 17.

Point 2: The introduction section can be presented in another way. The authors can give the introduction section in some terms like the general idea that already demonstrates an interest these days, the particular domain that uses the current application, the main problem in this research, and the central gap founded by the authors in this paper. Some related works support the main claim in this paper and support this work by focusing on some issues. At the end of the introduction section, the authors should be given a clear and comprehensive paragraph to show the readers how this research has been done (the main problem, contributions, experiments, comparisons, results, and so on).

Response 2: Thanks for the reviewer’s constructive and valuable suggestion for the improvement of this manuscript. We have added the summary at the end of the introduction to introduce our main idea, contributions, experiments and results. We have revised some grammatical mistakes and reorganized some sentences at the introduction.

Detailed revisions as follows:

  1. We have added the summary at the end of the introduction. The summary we added on Line 111 is as follows:

“In sum, for the existing quantum-inspired models are rarely based on complex-valued neural networks, we proposed a new quantum-inspired fully complex-valued neural network, ComplexQNN. Our contributions are summarized as follows:

  1. Based on quantum computation theory and complex-valued neural networks, we propose the theory and architecture of the ComplexQNN.
  2. We introduce the details modules of the ComplexQNN fully based on complex-valued neural networks, including a complex-valued embedding layer, a quantum encoding layer, and a measurement layer.
  3. The ComplexQNN is evaluated with six sentiment classification datasets, including binary classification and multi-classification. We adopt two metrics accuracy and F1-score to evaluate our model and compare with five classical neural models (TextCNN, GRU, ELMo, BERT, and RoBERTa). The experimental results show that the ComplexQNN has improved by 10% in accuracy metric compared with TextCNN and GRU, and has competitive experimental results with ELMo, BERT, and RoBERTa.”
  1. The expression “However, the problem of gradient explosion and gradient disappearance of RNNs make it difficult to build deep neural network.” has been revised into “But gradient explosion and gradient disappearance are problems with RNNs that make it hard to build a deep neural network.” on Line 47.
  2. The expression “These pre-training models can be used as text encoders, which can be easily adapted to different downstream tasks by fine-tuning strategies.” has been revised into “By using fine-tuning techniques, these pre-training models can serve as text encoders that are easily adaptable to various downstream tasks.” on Line 54.
  3. The expression “It has been proved that in some tasks, quantum computers have exponential computational complexity acceleration advantage.” has been revised into “It has been demonstrated that quantum computers have an exponential computational complexity acceleration advantage for some tasks.” on Line 63.
  4. The expression “Interestingly, some researchers build quantum-inspired models based on the mathematical framework of quantum computing theory.” has been revised into “Some researchers use the math behind quantum computing theory to build models that look like quantum computers.” on Line 82.
  5. The expression “A deep complex-valued convolutional network is proposed by Chiheb Trabelsi et al. [15] and has achieved” has been revised into “Trabelsi et al. [15] proposed a deep complex-valued convolutional network that has demonstrated” on Line 96.

Point 3: Some shortcuts word need to be justified like NISQ.

Response 3: We are grateful to the reviewer for this valuable and instructive comment. We have carefully checked the shortcuts word used in our manuscript and revised those incorrect words.

Detailed revisions as follows:

  1. We have revised the “NISQ” to “noisy intermediate-scale quantum (NISQ)” on Line 30.
  2. We have revised the “VGG” to “Visual Geometry Group (VGG)” on Line 248.
  3. We have revised the “ResNet” to “Residual Network (ResNet)” on Line 248.
  4. We have revised the “Yolo” to “You Only Look Once (YOLO)” on Line 249.
  5. We have updated the abbreviations used in our manuscript on Line 687, including ELMo, BERT, RoBERTa, CR, MPQA, MR, SST, SUBJ, VGG, ResNet, YOLO, CTQW, CNM, TextTN, NNQLM, GTN, DTN, ICWE, CICWE, and ComplexQNN.

Point 4: A few of the references in Related Work are old. Authors need to refer to recent publications in this area.

Response 4: Thanks for the reviewer’s constructive and valuable suggestion for the improvement of this manuscript. We have retained two classical algorithms in quantum computing, Shor and Grover algorithm. And we have added some recent publications on quantum computation. Thank you for pointing out the problem in our manuscript.

Detailed revisions as follows:

  1. We have added “Wang et al. [18] proposed in 2021 a quantum AdaBoost algorithm with a quadratic speedup. Apers et al. [19] proposed in 2022 a continuous-time quantum walks (CTQWs) search algorithm, which achieves a general quadratic speedup over classical random walks on an arbitrary graph. Huang et al. [20] proposed a quantum principal component analysis that achieved almost four orders of magnitude of reduction over the best-known classical lower bounds.” on Line 142.
  2. We have removed “David Deutsch first proposed in 1984 the concept of a universal quantum computer [16].” on Line 137.

Point 5: The authors should give the readers a straightforward method of choosing the experiments and their design. This will help the future researcher in this domain conduct new research and start from the current paper.

Response 5: Thanks for the reviewer’s instructive comments. To help the future researcher quickly learn about this work and this domain, we open source our code, and our code is available at Github: https://github.com/Levyya/ComplexQNN.

Point 6: The discussion of the results needs to include the strengths and weaknesses of the proposed algorithm.

Response 6: Thanks for the reviewer’s constructive and valuable suggestion for the improvement of this manuscript. We have added the weaknesses on Subsection 4.3 disscussion.

Detailed revisions as follows:

  1. We have added disscussion “However, we still face some challenges to overcome. First, this algorithm uses classical neural networks, including complex-valued neural networks, to simulate the quantum computing process. Therefore, additional parameters are required to represent the imaginary parts. Second, the complex-valued network space is more complex than the real-number space, and the design of complex-valued networks, such as optimizers and network modules, needs further research. Third, the scale of existing quantum computers are too small and too expensive to use. It is difficult to completely migrate the existing quantum-inspired methods to real quantum computers.” into Subsection 4.3 on Line 644.

In brief, we have tried our best to revise and improve the manuscript and made great changes in the manuscript according to the editors’ and reviewers’ constructive and thoughtful comments, and we hope that the corrections will meet with approval. Once again, we would like to thank all editors and reviewers very much for their valuable comments and suggestions that greatly help to improve the presentation of the paper.

Reviewer 3 Report

Comments in attached file.

Author Response

Dear Reviewer

In regards to the following

Article Number: axioms-2260415

“Quantum-Inspired Fully Complex-Valued Neutral Network for Sentiment Analysis”

Thank you very much for your instructive and constructive comments.

Those comments are all valuable and very helpful for revising and improving our paper, and with the important guiding significance to our researches. We have tried our best to revise and improve the manuscript according to the reviewers’ thoughtful and constructive comments. The manuscript has now been duly revised by taking into account all comments made by both editors and reviewers, and special changes are listed on the subsequent pages of this letter. Electronic versions of both this letter and the revised manuscript are included as well.

We appreciate for editors’ and reviewers’ work earnestly, and hope that the corrections will meet with approval. Once again, thank you very much for your comments and suggestions. We look forward to your information about our revised manuscript.

In closing, we would like to thank you for the much time and effort in handling our manuscript and in providing helpful improvement suggestions.

With best regards,

Yours sincerely,

Wei Lai

Jin-jing Shi

Yan Chang

School of Computer Science and Engineering,

Central South University,

Changsha, 410083, China

Response to Reviewer 3 Comments

We would like to thank you for very constructive and valuable comments which have great guiding significance for our researches.

Point 0: This is a very interesting and well written paper. In addition to the main research goal authors provided a well written introduction to quantum based computing.

The approach proposed is a sort of quantum-ready algorithm. It has a potential when quantum hardware becomes available, but is very difficult to predict when it happens. As things and now the algorithm is very greedy on storage, although it seems it offers slightly better accuracy.

It would be interesting if you could provide

Point 1: one more table showing actual time needed for learning and classification

Response 1: Thanks for the reviewer’s instructive comments. We retrain our code and record the training time on an epoch and create a table to show the training time for six models on six datasets.

Detailed revisions as follows:

  1. We have added the table 4 to show the training time on Page 18.

Point 2: some comments on what types of words/problems are most difficult for your approach.

Response 2: We are grateful to the reviewer for this valuable and instructive comment. We have added some comments on disscussion to introduce the problems that are difficult for our approach.

Detailed revisions as follows:

  1. We have added a disscussion “However, we still face some challenges to overcome. First, this algorithm uses classical neural networks, including complex-valued neural networks, to simulate the quantum computing process. Therefore, additional parameters are required to represent the imaginary parts. Second, the complex-valued network space is more complex than the real-number space, and the design of complex-valued networks, such as optimizers and network modules, needs further research. Third, the scale of existing quantum computers are too small and too expensive to use. It is difficult to completely migrate the existing quantum-inspired methods to real quantum computers.” into Subsection 4.3 on Line 652.

Point 3: Is the algorithm/implementation publiclly available?

Response 3: We are grateful to the reviewer for this valuable and instructive comment. Our code is available at Github: https://github.com/Levyya/ComplexQNN.

Point 4: Line 125 where are all real numbers, something is missing here

Response 4: Thanks for the reviewer’s instructive comments. We feel sorry for our carelessness. We have added the missing symbol in this sentence. Thank you so much for your careful check, and the mistakes have been corrected in the revised manuscript.

Detailed revisions as follows:

  1. We have revised the sentence to “where \added{$\gamma, \theta, \varphi$} are all real numbers” on Line 161.

Point 5: L. 140 is consists should be -> consists

Response 5: We sincerely appreciate the reviewer’s reminder. We have deleted the extra “is”. Thank you so much for your careful check, and the mistakes have been corrected in the revised manuscript.

Detailed revisions as follows:

  1. We have deleted the extra “is” on Line 176.

Point 6: L. 168 the sentence "One is the need to close the classical intuition, and the algorithm based 168 on intuition can only think of classical algorithms; the other is that the designed quantum 169 algorithm must exceed all known classical algorithms" should be revised as now it is difficult to understand

Response 6: Thanks for the reviewer’s constructive and valuable suggestion for the improvement of this manuscript. We feel sorry for not describing this sentence clearly. Here we want to describe why it is so difficult to come up with a new quantum algorithm. According to the reviewer’s comment, we have reorganized the sentece.

Detailed revisions as follows:

  1. The sentence has been revised into “First, quantum computing uses qubits instead of classical bits, so quantum algorithms need to consider how to use the superposition and entanglement properties of qubits to achieve parallel computing. Second, quantum algorithms need to be more efficient than existing classical algorithms. Otherwise there is no need to use a quantum computer. This is difficult to achieve with current hardware constraints, and most quantum algorithms can only be theoretically proven to have an acceleration advantage.” on Line 206.

Point 7: L.180 compl ex-valued -> complex-valued

Response 7: We sincerely appreciate your comments. We have changed the “compl ex-valued” to “complex-valued”. We feel sorry for our carelessness. Thank you so much for your careful check, and the mistakes have been corrected in the revised manuscript.

Detailed revisions as follows:

  1. We have changed the “compl ex-valued” to “complex-valued” on Line 224.

In brief, we have tried our best to revise and improve the manuscript and made great changes in the manuscript according to the editors’ and reviewers’ constructive and thoughtful comments, and we hope that the corrections will meet with approval. Once again, we would like to thank all editors and reviewers very much for their valuable comments and suggestions that greatly help to improve the presentation of the paper.
